# Development and validation of an LC-MS/MS method for determination of B vitamins and some its derivatives in whole blood

**David Kahoun**[1]*, **Pavla Fojtíková**[1], **František Vácha**[1], **Marie Čížková**[1], **Roman Vodička**[2], **Eva Nováková**[3], **Václav Hypša**[3]

**1** Department of Chemistry, Faculty of Science, University of South Bohemia in České Budějovice, České Budějovice, Czech Republic, **2** Prague Zoo, Prague, Czech Republic, **3** Department of Parasitology, Faculty of Science, University of South Bohemia in České Budějovice, České Budějovice, Czech Republic

* dkahoun@prf.jcu.cz

**Data Availability Statement:** All relevant data are within the manuscript and its Supporting Information files.

## Abstract

Obligate symbiotic bacteria associated with the insects feeding exclusively on vertebrate blood are supposed to complement B vitamins presumably lacking in their diet. Recent genomic analyses revealed considerable differences in biosynthetic capacities across different symbionts, suggesting that levels of B vitamins may vary across different vertebrate hosts. However, a rigorous determination of B vitamins content in blood of various vertebrates has not yet been approached. A reliable analytical method focused on B vitamin complex in blood can provide valuable informative background and understanding of general principles of insect symbiosis. In this work, a chromatographic separation of eight B vitamins (thiamine, riboflavin, niacin, pantothenic acid, pyridoxine, biotin, folic acid, and cyanocobalamine), four B vitamin derivatives (niacinamide, pyridoxal-5-phosphate, 4-pyridoxic acid, and tetrahydrofolic acid), and 3 stable isotope labelled internal standards was developed. Detection was carried out using dual-pressure linear ion trap mass spectrometer in FullScan MS/MS and SIM mode. Except for vitamin B9 (tetrahydrofolic acid), the instrument quantitation limits of all analytes were ranging from 0.42 to 5.0 μg/L, correlation coefficients from 0.9997 to 1.0000, and QC coefficients from 0.53 to 3.2%. Optimization of whole blood sample preparation step was focused especially on evaluation of two types of protein-precipitation agents: trichloroacetic acid and zinc sulphate in methanol. The best results were obtained for zinc sulphate in methanol, but only nine analytes were successfully validated. Accuracy of the procedure using this protein-precipitating agent was ranging from 89 to 120%, precision from 0.5 to 13%, and process efficiency from 65 to 108%. The content of B vitamins in whole blood samples from human and various vertebrates is presented as an application example of this newly developed method.

**Funding:** This research was supported by the Czech Science Foundation (website: https://gacr.cz/en/) under grant agreement No. 18-07711S. The funder had no role in study design, data collection and analysis, decision to publish, or preparation of the manuscript.

**Competing interests:** The authors have declared that no competing interests exist.

## Introduction

B vitamins are supposed to play a central role in the origin and evolution of obligate symbiosis between insects and bacteria. It has been generally accepted that the main role of the maternally inherited obligate symbionts, so called primary symbionts (P-symbionts), is to provide their hosts with compounds missing in their diets, mainly amino acids, vitamins, and cofactors [1]. Two ecological groups of insects are recognized as typical P-symbiont hosts, namely phytophages (e.g. aphids with *Buchnera*) and hematophages (e.g. tsetse with *Wigglesworthia*). The early experiments on the tsetse-*Wigglesworthia* system showed that elimination of symbionts by antibiotics causes loss of fertility which can be restored by dietary supplementation with B vitamins [2]. Similar results were reported for human louse *Pediculus humanus* and its bacterial symbionts [3]. Based on these experiments, it was hypothesized that biosynthesis of B vitamins is one of the main roles of these P-symbionts. The molecular era then brought proofs of genome capacity for B vitamin synthesis in *Wigglesworthia glossinidia* and also some other P-symbionts in blood feeding insects [4–6]. The lack of B vitamins in vertebrate blood and their provision by P-symbionts thus became a generally accepted reason for the symbiosis between blood feeding insects and bacteria.

Recently, genomic data became available for several symbionts of hematophagous hosts. Their overview indicates that the biosynthetic capacities vary across different insects-bacteria associations and different vitamins. For example, biotin seems to be mostly provided by the symbiotic bacteria and several instances of horizontal gene transfer of a complete biotin operon into the symbionts from unrelated bacteria suggest a crucial role of this pathway in the host- symbiont interaction [5, 7]. On the other hand, many symbionts in blood-feeding insects lack the capacity to produce thiamine and they may even scavenge this compound from the host [5, 8].

Considering these long-lasting debates and the significance of blood feeding insects, we found it striking how very little is known about B vitamin content in blood of different vertebrate species. This example illustrates an importance of a precise analytical assay of B vitamins in blood when addressing particular biological questions.

Vitamins of the B complex (B1 –thiamine, B2 –riboflavin, B3 –niacin, B5 –pantothenic acid, B6 –pyridoxine, B7 –biotin, B9 –folic acid and B12 –cyanocobalamine) are a set of low-molecular-weight water-soluble substances that play a key role in many metabolic pathways. Vertebrates generally maintain a low concentration of B vitamins in the blood, do not store them, and remain dependent on their dietary content.

Some of the B vitamins can be found in blood or tissues in different forms, i.e. vitamers or provitamins. Vitamin B1 may be present as the thiamine monophosphate, diphosphate or triphosphate esters, vitamin B3 as the niacinamide, vitamin B6 as the aldehyde pyridoxal, amine pyridoxamine, their corresponding 5′-phosphate derivatives, and the pyridoxic acid as a degradation product, vitamin B9 as the tetrahydrofolic acid, and vitamin B12 as methyl-, hydroxo- or adenosyl- derivate of cobalamin.

Levels of individual B vitamins are routinely measured in a clinical practice and physiology research. However, each vitamin of the B complex is either analysed separately [9, 10] or, according to the individual study needs, a few are determined simultaneously [11, 12]. Various methods of B vitamins analysis are generally used, including microbiological assays, radio immunoassay, protein binding, spectrophotometry, fluorimetry, chemiluminescence, capillary electrophoresis, or high-performance liquid chromatography (HPLC) [13–15]. However, using different analytical methods for different B vitamins introduces many obstacles into the comprehensive surveys where comparing precise levels is crucial.

Recently, liquid chromatography has been established as the most frequently used emerging method for the determination of B vitamins, with the advantage of multi-analytes detection of

most of the water-soluble vitamins in a single run. The tandem mass spectrometry (MS/MS) detection shows the best results in sensitivity and specificity. Several methods for simultaneous detection of some B vitamins using LC-MS/MS are reported in literature for the analysis of various food matrices, human milk, plasma or whole blood [10, 16–25].

Unfortunately, a comprehensive method for the analysis of all B vitamins in whole blood samples is missing. With respect to blood as a nutrient source for hematophagous arthropods and the fact that about 80 to 90% of B1 vitamin is present within red blood cells [26–28], we present a comprehensive analytical method for separation and detection of B vitamins from a whole blood sample in a single run based on LC-MS/MS.

## Materials and methods

### Chemicals and reagents

All standards and internal standards (IS) were purchased from Sigma-Aldrich: thiamine hydrochloride (CAS: 67-03-8, purity: CRM, abr.: B1), riboflavin (CAS: 83-88-5, purity: CRM, abr.: B2), niacin (CAS: 59-67-6, purity: CRM, abr.: B3), niacinamide (CAS: 98-92-0, abr.: purity: CRM, B3-AM), calcium-d-pantothenate (CAS: 137-08-6, purity: CRM, abr.: B5), pyridoxine hydrochloride (CAS: 58-56-0, purity: CRM, abr.: B6), 4-pyridoxic acid (CAS: 82-82-6, purity: ≥98%, abr.: B6-4PA), pyridoxal 5′-phosphate hydrate (CAS: 853645-22-4, purity: ≥98%, abr.: B6-5P), biotin (CAS: 58-85-5, purity: CRM, abr.: B7), folic acid (CAS: 59-30-3, purity: CRM, abr.: B9), tetrahydrofolic acid (CAS: 135-16-0, purity: ≥65%, abr. B9-THF), cyanocobalamine (CAS: 68-19-9, purity: CRM, abr.: B12), nicotinic acid-$^{13}C_6$ (CAS: 1189954-79-7, 100 μg/mL in methanol, abr. IS-B3), calcium pantothenate-$^{13}C_6$, $^{15}N_2$ (di-β-alanine-$^{13}C_6$, $^{15}N_2$) (CAS: N.A., purity ≥97%, abr.: IS-B5), and biotin-(ring-6,6-d2) (CAS: 1217481-41-8, purity ≥97%, abr.: IS-B7). The compounds were stored in the original packages at -75°C prior to use.

Acetonitrile (LiChrosolv, hypergrade for LC-MS), methanol (LiChrosolv, hypergrade for LC-MS), formic acid (eluent additive for LC-MS), ammonium formate (eluent additive for LC-MS), zinc sulfate heptahydrate (ACS reag. ≥99%), and trichloroacetic acid (ACS reag. ≥99.5%) were also purchased from Sigma-Aldrich. Deionized water was prepared using a purification system Thermo Scientific Smart2Pure 6 UV/UF.

### Standard, internal standard and calibration solutions

Standard stock solutions were prepared individually by dissolving an appropriate amount of each compound in methanol (B1, B2, B3, B3-AM, B6 and B12), in 1% HCOOH (B5), in 0.5% NH$_4$OH (B7), in 4% NH$_4$OH (B9), in 0.01M HCl (B6-4PA), in 50% methanol (B6-5P) or in water (B9-THF) to obtain concentration approx. 1000 mg/L (except for B2, B6-4PA, B6-5p and B9-THF where the concentration was 10 mg/L). All standard stock solutions were stored at -75°C. The standard mixture solution A (10 mg/L) was prepared daily in a glass volumetric flask (5 mL) by diluting the calculated amount of each standard stock solution with water and then diluted to obtain the standard mixture working solution B (1 mg/L).

The internal standard stock solutions were prepared individually by dissolving an appropriate amount of each compound in 1% HCOOH (IS-B5) or in 0.5% NH$_4$OH (IS-B7) to obtain concentration approx. 200–500 mg/L. The stock solution of the internal standard IS-B3 was obtained commercially as an ampule of 1 mL solution (100 mg/L in methanol). All internal standard stock solutions were stored at -75°C. The internal standard working mixture solution was prepared daily by diluting of 100 μl of each internal standard stock solution with 4.7 mL of water to obtain concentration 2–5 mg/L.

The calibration solutions were prepared daily in brown glass HPLC crimp vials (1.8 mL) diluting the standard working solution B (5–500 μL) and the internal standard working solution (100 μL) with water (400–895 μL) to obtain 9 different concentrations: 1; 2.5; 5; 10; 25; 50; 100; 250 and 500 μg/L. Total volume of each calibration solution was 1000 μL. The calibration solutions were analysed in triplicates.

## Sample collection and preparation

The whole blood samples used for the method development and validation were collected from a human volunteer by a licensed practical nurse into 3 mL blood collection tubes containing an anticoagulant potassium ethylenediaminetetra-acetic acid (K3EDTA) purchased from Dialab. The blood collection tubes were mixed and immediately placed on ice. The samples were prepared for the analysis as described below within an hour after the sample collection and immediately analysed in technical triplicates. The use of human blood samples was approved by the Ethical Committee of the University of South Bohemia. The participant's informed written consent was provided.

The animal whole blood samples were collected similarly either during regular health inspection or a medical intervention, usually in the morning after overnight starvation period. The samples were kept on ice until frozen in liquid nitrogen or deep freezer at -75˚C.

**Sample preparation using TCA.** For the analysis using trichloroacetic acid as a precipitating agent (TCA), 100 μL of water and 100 μL of the internal standard working solution were mixed in a 1.8 mL Eppendorf tube with 500 μL of whole blood and then deproteinized by adding 300 μL of TCA-agent (2.0 g of trichloroacetic acid in 8.0 mL of water). The samples were vortexed for ~10 s, placed on ice for 15 min and then centrifuged at 20,000xg for 10 min. Clarified supernatant was transferred to the brown glass HPLC crimp vial (1.8 mL) for further analysis. The spiked samples were prepared as mentioned above and 100 μL of an appropriate standard solution was used instead of 100 μL water. In the case of the blank samples, 500 μL of water was used instead of 500 μL of whole blood.

**Sample preparation using $ZnSO_4$/methanol.** The samples with precipitating agent zinc sulfate heptahydrate in methanol were prepared and analysed. Water (100 μL) and the internal standard working solution (100 μL) were mixed in a 1.8 mL Eppendorf tube with 300 μL of whole blood and then deproteinized by adding 500 μL of $ZnSO_4$-agent (300mM $ZnSO_4.7H_2O$ mixed with methanol in a ratio of 3:7 v/v). The samples were vortexed for ~10 s, placed on ice for 15 min and then centrifuged at 20,000xg for 10 min. Clarified supernatant was transferred to the brown glass HPLC crimp vial (1.8 mL) for further analysis. The spiked sample were prepared as mentioned above. Blank samples contained 300 μL of water instead of 300 μL of whole blood.

## LC-MS/MS analysis

Liquid chromatography was performed by a Thermo Scientific Dionex Ultimate 3000. Quaternary Analytical system was consisted of a solvent rack SRD-3600, binary pump HPG-3400RS, autosampler WPS-3000TRS, column compartment TCC-3000RS, diode array detector DAD-3000RS, heated electrospray HESI II probe, mass detector Velos Pro, and a PC with LTQ Tune Plus 2.8 and Xcalibur 4.2.47 software.

The LC-MS/MS system was equipped with a chromatographic column Thermo Acclaim C30 (150 x 2.1 mm, 3 μm) maintained at 15˚C. Mobile phase A was prepared by diluting 0.6306 g of ammonium formate with 1 L of water containing 175 μL of formic acid (pH 4.0). Mobile phase B was prepared by diluting 0.6306 g of ammonium formate with 1 L of water containing 1750 μL of formic acid (pH 3.0). Mobile phase C was prepared by diluting 0.0568 g

of ammonium formate with 10 mL of mobile phase B and 90 mL of acetonitrile. All mobile phases were prepared daily, filtered through a 0.2 μm nylon filter and degassed before use.

Chromatographic separation was achieved at a flow rate 0.4 mL/min using a gradient of mobile phase A, B and C as follows: (I) 100% mobile phase A from 0 to 5 min, (II) 100% mobile phase B from in 5 min, (III) linear gradient increase to 35% C from 5 to 17 min, (IV) equilibration at starting conditions (100% A) from 17 to 25 min. Injection volume was 10 μl.

Mass spectrometry analysis was completed by multiple reaction monitoring (MRM) or single ion monitoring (SIM) in ESI positive mode using the following tune parameters: capillary voltage = 4 kV, desolvation temperature = 350 ˚C, sheath gas flow rate = 60 arb., auxiliary gas flow rate = 20 arb., transfer capillary temperature = 350˚C, S-lens RF level = 60%, Front lens = -9 V, Ion time (MRM) = 200 ms, Ion time (SIM) = 200 ms.

## Method validation

Linearity of calibration curves constructed from 6–9 concentration levels (each concentration level in triplicate) was evaluated using two acceptability criteria: correlation coefficient ($R \geq 0.9900$) and quality coefficient ($QC \geq 5.0\%$). Instrument detection limit (IDL) and instrument quantitation limit (IQL) were calculated based on signal-to-noise ratio $S/N = 3$ and $S/N = 10$, respectively.

Accuracy, precision and process efficiency were assessed using spiked whole blood samples to achieve a concentration increase of all analytes of 10 and 100 μg/L. All spiked samples were prepared by both sample preparation procedures using TCA and $ZnSO_4$/methanol, each in triplicate. Accuracy was determined as the relative error (%) and was expressed as recovery: (measured concentration—nominal concentration/nominal concentration) × 100. The precision was determined as the coefficient of variation (%) and was expressed as repeatability: (standard deviation/mean concentration measured) x 100. Process efficiency (%) was determined: (peak area of an analyte in spiked sample/peak area of the same analyte in standard solution) x 100.

Potential carry-over effects were assessed by monitoring the signal in a blank sample analysed immediately after the analysis of the highest calibration solution (500 μg/L).

## Results and discussion

### Optimization of chromatographic and detection conditions

The initial chromatographic and detection conditions were taken over from the Thermo Application Note 294 [29]. The method was optimized to achieve the most stable detector response and the lowest limits of quantitation as possible. At first, ion source parameters were tested to prove that stability of detector response is less than 15%. Then, fundamental ion optic parameters S-lens RF level (from 0 to 70%; optimum at 60%) and Front lens voltage (from -15.0 to -5 V; optimum at -9.0 V) were optimized to ensure the most effective ion transmission from the ion source to the mass analyser. Finally, normalised collision energy (from 0 to 100%), product ion choice and ion time (from 1 to 500 ms) were chosen to achieve the highest detector response and sufficient number of data points (at least 15 points) across a chromatographic peak. Optimal normalised collision energy needed to achieve optimum fragmentation efficiency ranged from 25 to 50% for almost all analytes excluding B3, B3-AM and IS-B3. Precursor ions of these compounds do not provide any stable product ion after application of any normalised collision energy–precursor ions either do not fragment at all or fragment totally using this linear ion trap mass analyser. Due to this fact, only SIM mode remained to be chosen in this case. With respect to relatively low scan speed (in comparison with e.g., triple quadrupole or time-of-flight mass analysers), mass detection was divided into

**Table 1. Optimal detection conditions.**

| Compound | Class | Retention time | Scan window | SIM | MRM | | Normalized collision energy |
|---|---|---|---|---|---|---|---|
| | | | | | Precursor ion | Production | |
| | | (min) | (min) | (1) | (1) | (1) | (%) |
| IS—Vitamin B3 (Niacin) | Internal standard | 2.6 | 2.0–7.0 | 130.0 | - | - | - |
| Vitamin B3 (Niacin) | Analyte | 2.6 | | 124.0 | - | - | - |
| Vitamin B1 (Thiamine) | Analyte | 4.4 | | - | 265.1 | 122.0 | 30 |
| Vitamin B6-5P (Pyridoxal 5-phosphate) | Analyte | 6.6 | 5.0–11.0 | - | 248.1 | 150.0 | 27 |
| Vitamin B6 (Pyridoxine) | Analyte | 7.4 | | - | 170.1 | 152.0 | 30 |
| Vitamin B3-AM (Niacinamide) | Analyte | 7.7 | | 123 | - | - | - |
| IS—Vitamin B5 (Pantothenic acid) | Internal standard | 13.7 | 11.0–15.5 | - | 224.1 | 206.0 | 50 |
| Vitamin B5 (Pantothenic acid) | Analyte | 13.7 | | - | 220.1 | 202.0 | 50 |
| Vitamin B6-4PA (4-Pyridoxic acid) | Analyte | 13.8 | | - | 184.1 | 166.0 | 25 |
| Vitamin B9-THF (Tetrahydrofolic acid) | Analyte | 14.6 | | - | 446.2 | 299.1 | 45 |
| Vitamin B12 (Cyanocobalamine) | Analyte | 15.6 | 15.0–20.0 | - | 678.4 | 359.0 | 20 |
| Vitamin B9 (Folic acid) | Analyte | 15.7 | | - | 442.2 | 295.0 | 50 |
| IS—Vitamin B7 (Biotin) | Internal standard | 16.4 | | - | 247.1 | 229.0 | 50 |
| Vitamin B7 (Biotin) | Analyte | 16.4 | | - | 245.1 | 227.0 | 50 |
| Vitamin B2 (Riboflavin) | Analyte | 16.5 | | - | 377.1 | 243.0 | 30 |

4 different scan windows (2.0–7.0 min, 5.0–11.0 min, 11.0–15.5 min and 15.0–20.0 min) during chromatographic analysis to ensure as long ion time for each analyte or internal standard as possible. All optimal detection conditions for each analyte are listed in Table 1, general detection conditions are mentioned at the end of Chapter LC-MS/MS analysis and representative chromatogram is shown in Fig 1.

## Validation characteristics of calibration curves

The calibration curves were measured within the range of 1–500 μg/L, but the whole range of the acceptability criteria (R ≥ 0.9900 and QC ≥ 5.0%) were fulfilled only for B2, B6, B7 and B12. For other analytes, calibration ranges needed to be narrowed down either from the lowest concentrations (B3, B3-AM, B5 and B6-5P) of from the highest concentrations (B1, B6-4PA or B9). The calibration curve of B9-THF was not assessed due the lack of data caused by an insufficient detector response–instrument quantitation limit lies at the highest calibration level (500 μg/L). The correlation coefficients ranged within very good interval of 0.9997–1.0000.

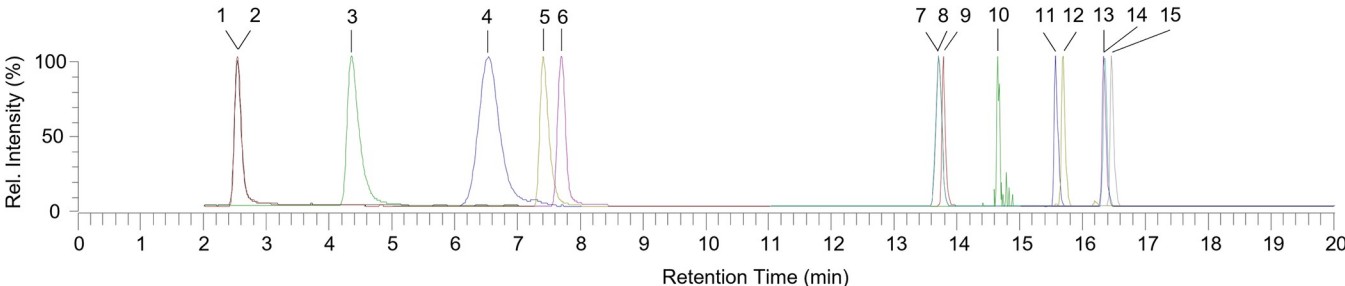

**Fig 1. Chromatogram of B vitamins standards and internal standards at (500 μg/L).** Peak identification: 1—IS-B3 (Stable isotope labelled internal standard), 2—B3 (Niacin), 3—B1 (Thiamine), 4 - B6-5P (Pyridoxal 5-phosphate), 5—B6 (Pyridoxine), 6 - B3-AM (Niacinamide), 7—IS-B5 (Stable isotope labelled internal standard), 8—B5 (Pantothenic acid), 9 - B6-4PA (4-Pyridoxic acid), 10 - B9-THF (Tetrahydrofolic acid), 11—B12 (Cyanocobalamine), 12—B9 (Folic acid), 13—IS-B7 (Stable isotope labelled internal standard), 14—B7 (Biotin), 15—B2 (Riboflavin).

**Table 2. Validation characteristics of calibration curves.**

| Analyte | Internal standard | Instrument detection limit | Instrument quantitation limit | Range | Correlation coefficient | Quality coefficient |
|---|---|---|---|---|---|---|
| | (-) | (µg/L) | (µg/L) | (µg/L) | (1) | (%) |
| Vitamin B1 (Thiamine) | IS-B3 | 0.13 | 0.42 | 1–100 | 0.9999 | 2.0 |
| Vitamin B2 (Riboflavin) | IS-B7 | 0.27 | 0.91 | 1–500 | 1.0000 | 1.3 |
| Vitamin B3 (Niacin) | IS-B3 | 1.2 | 3.8 | 10–500 | 0.9999 | 1.7 |
| Vitamin B3-AM (Niacinamide) | IS-B3 | 1.5 | 5.0 | 5–500 | 0.9998 | 2.7 |
| Vitamin B5 (Pantothenic acid) | IS-B5 | 0.18 | 0.56 | 2.5–500 | 1.0000 | 0.53 |
| Vitamin B6 (Pyridoxine) | IS-B5 | 0.14 | 0.45 | 1–500 | 0.9998 | 2.9 |
| Vitamin B6-5P (Pyridoxal 5-phosphate) | IS-B3 | 1.4 | 4.5 | 5–500 | 0.9999 | 1.8 |
| Vitamin B6-4PA (4-Pyridoxic acid) | IS-B5 | 0.30 | 1.0 | 1–100 | 0.9997 | 3.2 |
| Vitamin B7 (Biotin) | IS-B7 | 0.27 | 0.91 | 1–500 | 1.0000 | 1.3 |
| Vitamin B9 (Folic acid) | IS-B7 | 0.21 | 0.71 | 1–100 | 0.9997 | 3.4 |
| Vitamin B9-THF (Tetrahydrofolic acid) | IS-B7 | 150 | 500 | n.a. | n.a. | n.a. |
| Vitamin B12 (Cyanocobalamine) | IS-B5 | 0.30 | 1.0 | 1–500 | 0.9999 | 2.2 |

The quality coefficients ranged from 0.53 to 3.4%. The best validation characteristics were obtained for B5 (R = 1.0000, QC = 0.53%), the worst results were achieved for B9 (R = 0.9997, QC = 3.4%). The instrument quantitation limits ranged from 0.42 to 4.5 µg/L except for B9-THF (500 µg/L). The validation data are summarized in Table 2, a detailed evaluation of the calibration curves (including graphs) is given in S1 File.

## Effect of protein-precipitating agents on accuracy, precision and process efficiency of the method

Protein-precipitating agents could play a crucial role in sample pre-treatment for trace analysis. Whole blood is one of the most challenging matrixes for LC/MS technique due to negative ionization effects during electrospray ionization. A comprehensive study [30] focused on optimization of protein precipitation from plasma samples, based upon effectiveness of protein removal and ionization effect in LC-MS/MS, shows that 10% TCA is an optimal protein-precipitating agent for removal of plasma proteins, together with minimal ionization effect. Nevertheless, whole blood is more complex matrix than plasma and the sample pre-treatment step must overcome not only sufficient protein precipitation but also haemolysis step allowing for an efficient release of analytes presented in erythrocytes. Finally, physicochemical properties (e.g., pH) of a protein-precipitating agent have to be taken into consideration with regard to the analytes stability. One of the very effective non-acidic agents for whole blood precipitation is 0.2 M zinc sulphate solution in an organic solvent [31, 32]. Both sample preparation methods mentioned above were assessed for presented determination of B vitamins in whole blood using LC/MS in this article.

Protein precipitation using $ZnSO_4$/methanol provides better results at both spike levels 10 and 100 µg/L in general, but B6-5P, B9 and B9-THF were not detected regardless of used protein-precipitating agent. The ranges of evaluated parameters (accuracy, precision and process efficiency) of the remaining analytes were narrower and closer to the optimal values in almost all cases except for the precision at 100 µg/L (1.0–13% for $ZnSO_4$/methanol, but 1.6–6.8% for TCA). Comparably worse precision in case of protein precipitation using $ZnSO_4$/methanol was observed for B1 at both spike levels, i.e.– 9.9% at 10 µg/L (TCA 4.0%) and 13% at 100 µg/L

(TCA 1.6%). This higher variation of precision values is caused by peak broadening due to the high percentage of the organic solvent in $ZnSO_4$/methanol solution (70%), relatively high injection volume (10 μL), and a narrow chromatographic column (2.1 mm). Excluding B1 results, precision ranged from 0.50 to 7.1% at 10 μg/L and from 1.0 to 6.2 at 100 μg/L.

The choice of the precipitating agent significantly affects the accuracy of the presented method. In the case of TCA agent the accuracy ranged from 26 to 122% at 10 μg/L and from 30 to 112% at 100 μg/L. Comparably better results were obtained in the case of $ZnSO_4$/methanol—from 90 to 114% at 10 μg/L and from 89 to 120% at 100 μg/L. Low accuracy when using TCA is noticeable for three B vitamins: B12 (26% at 10 μg/L and 30% at 100 μg/L), B2 (51% at 10 μg/L and 56% at 100 μg/L), and B5 (55% at 10 μg/L and 68% at 100 μg/L).

Process efficiency (PE) assesses LC-MS method as a whole, considering both matrix effects and the efficiency (recovery) of the extraction process. The method using TCA agent for protein precipitation step shows rather big differences at both spike levels 10 and 100 μg/L in this parameter–PE of three B vitamins were higher than 150% (B1, B3 and B6), PE of three B vitamins were lower than 50% (B2, B7 and B12) and only three B vitamins provides PE between 50–150% (B3-AM: 94 and 101%; B5: 111 and 117%; B6-4PA: 100 and 107%). Much better results were obtained using the $ZnSO_4$/methanol method for which PE were ranging from 65 to 108% at both spike levels 10 and 100 μg/L for all nine B vitamins (B1, B2, B3, B3-AM, B5, B6, B6-4PA, B7 and B12).

Based on a comprehensive comparison of all the results of testing both methods using trichloroacetic acid (Table 3, Figs 2 and 3) and $ZnSO_4$/methanol (Table 4, Figs 2 and 3), the $ZnSO_4$/methanol protein-precipitation agent was chosen for further work.

## Stability of B vitamins in whole blood samples

The stability of B vitamins in the whole blood samples spiked at 100 μg/L level was evaluated at -18°C to check whether a short-time storage of blood in collection tubes in an ordinary freezer does not affect concentration of the analytes. The stability of B6 is decreased at such conditions. 17% content decrease was observed after one day, and 51% content decrease in a four-day storage time. The content of B1, B2, B3, B5, B6-PA, B7 and B12 was stable for the first three days. On the fourth day, a decrease of 5–23% was observed. Such results correspond to the previously reported observations of stability of B vitamins in whole blood [33], B1 in the

**Table 3. Validation characteristics of the method using TCA as a precipitating agent.**

| Analyte | Spike 10 μg/L | | | Spike 100 μg/L | | |
|---|---|---|---|---|---|---|
| | Accuracy | Precision | Process efficiency | Accuracy | Precision | Process efficiency |
| | (%) | (%) | (%) | (%) | (%) | (%) |
| Vitamin B1 (Thiamine) | 73 | 4.0 | 229 | 78 | 1.6 | 166 |
| Vitamin B2 (Riboflavin) | 51 | 1.8 | 29 | 56 | 2.3 | 30 |
| Vitamin B3 (Niacin) | 91 | 8.4 | 250 | 102 | 5.7 | 199 |
| Vitamin B3-AM (Niacinamide) | 71 | 6.2 | 94 | 79 | 4.8 | 101 |
| Vitamin B5 (Pantothenic acid) | 55 | 2.7 | 117 | 68 | 1.6 | 111 |
| Vitamin B6 (Pyridoxine) | 122 | 7.1 | 156 | 112 | 4.7 | 151 |
| Vitamin B6-5P (Pyridoxal 5-phosphate) | < 1 | n.a. | n.a. | < 1 | n.a. | n.a. |
| Vitamin B6-4PA (4-Pyridoxic acid) | 88 | 5.8 | 100 | 85 | 2.2 | 107 |
| Vitamin B7 (Biotin) | 93 | 2.9 | 44 | 95 | 2.5 | 61 |
| Vitamin B9 (Folic acid) | < 1 | n.a. | n.a. | < 1 | n.a. | n.a. |
| Vitamin B9-THF (Tetrahydrofolic acid) | < 1 | n.a. | n.a. | < 1 | n.a. | n.a. |
| Vitamin B12 (Cyanocobalamine) | 26 | 24 | 41 | 30 | 6.8 | 43 |
| Range | 26–122 | 1.8–24 | 29–250 | 30–112 | 1.6–6.8 | 30–199 |

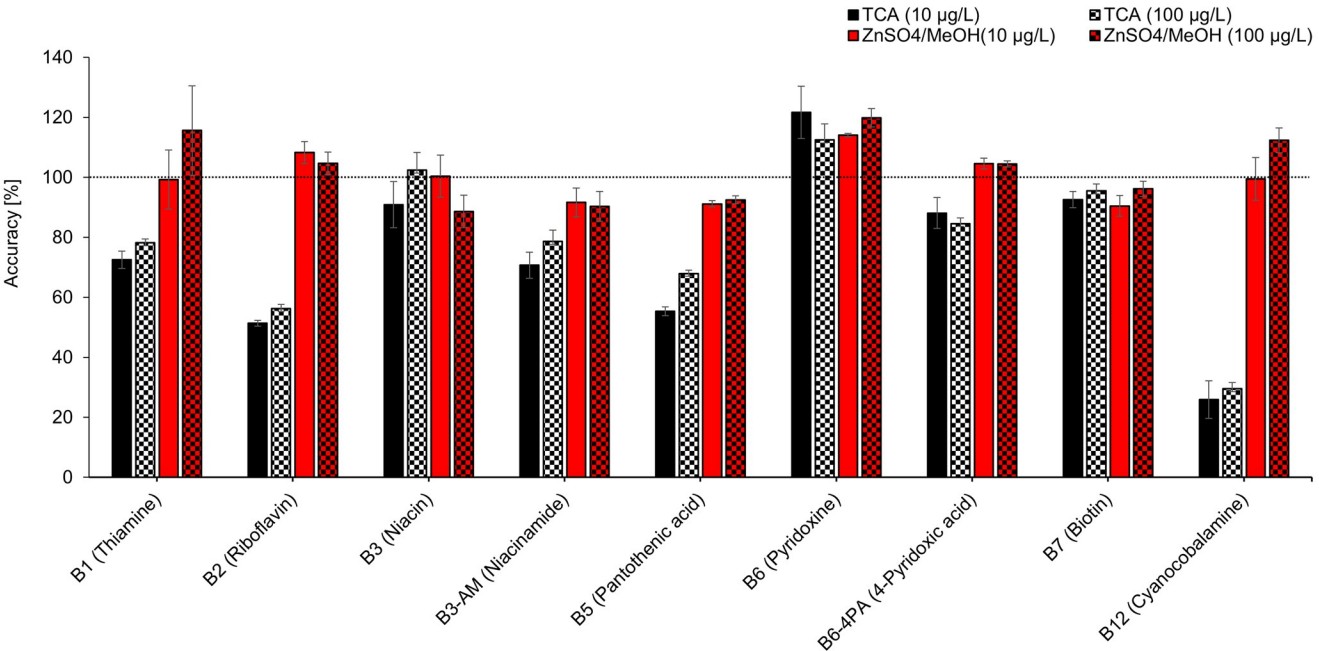

**Fig 2. Effect of precipitating agent on accuracy and precision.**

erythrocytes [28], or vitamins B9 and B12 in human serum [34]. Slight differences may be due to EDTA presence in the previous studies. The content of B3-AM could not be evaluated due to the high detector response that lies in the range of the calibration curve due to its extremely high concentration in the tested whole blood sample. However, B3-AM, as well as B3, are

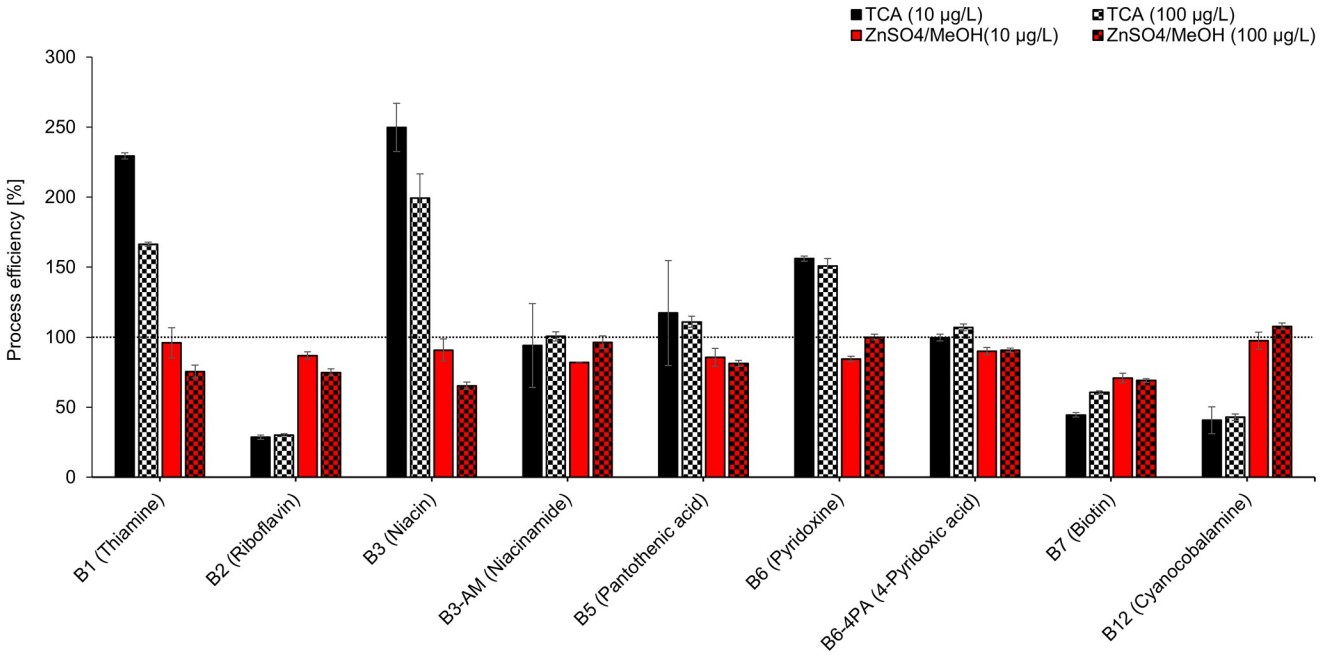

**Fig 3. Effect of precipitating agent on process efficiency.**

**Table 4. Validation characteristics of the method using ZnSO$_4$/methanol as a precipitating agent.**

| Analyte | Spike 10 µg/L | | | Spike 100 µg/L | | |
|---|---|---|---|---|---|---|
| | Accuracy | Precision | Process efficiency | Accuracy | Precision | Process efficiency |
| | (%) | (%) | (%) | (%) | (%) | (%) |
| Vitamin B1 (Thiamine) | 99 | 9.9 | 96 | 116 | 13 | 75 |
| Vitamin B2 (Riboflavin) | 108 | 3.4 | 87 | 105 | 3.6 | 75 |
| Vitamin B3 (Niacin) | 100 | 7.0 | 91 | 89 | 6.2 | 65 |
| Vitamin B3-AM (Niacinamide) | 92 | 5.3 | 82 | 90 | 5.5 | 96 |
| Vitamin B5 (Pantothenic acid) | 91 | 1.2 | 86 | 92 | 1.4 | 81 |
| Vitamin B6 (Pyridoxine) | 114 | 0.50 | 84 | 120 | 2.5 | 100 |
| Vitamin B6-5P (Pyridoxal 5-phosphate) | < 1 | n.a. | n.a. | < 1 | n.a. | n.a. |
| Vitamin B6-4PA (4-Pyridoxic acid) | 105 | 1.7 | 90 | 104 | 1.0 | 91 |
| Vitamin B7 (Biotin) | 90 | 3.8 | 71 | 96 | 2.6 | 69 |
| Vitamin B9 (Folic acid) | < 1 | n.a. | n.a. | < 1 | n.a. | n.a. |
| Vitamin B9-THF (Tetrahydrofolic acid) | < 1 | n.a. | n.a. | < 1 | n.a. | n.a. |
| Vitamin B12 (Cyanocobalamine) | 99 | 7.1 | 97 | 112 | 3.7 | 108 |
| Range | 90–114 | 0.50–9.9 | 71–97 | 89–120 | 1.0–13 | 65–108 |

generally known for a sufficient stability at various conditions and their activity is not affected by heat, light, acid, alkali, or oxidation [35]. Remaining B vitamins (B6-5P, B9 and B9-THF) were not detected due to unsuccessful validation characteristics of the tested sample preparation methods.

## Level of B vitamins in whole blood samples of various vertebrates

The new method was applied to a set of whole blood samples from various vertebrates, *Acipenser ruthenus*, *Alces alces*, *Arctocephalus pusillus*, *Bison bonasus*, *Camelus bactrianus*, *Dama dama*, *Delichon urbicum*, *Elephas maximus*, *Equus przewalskii*, *Fukomys mechowi*, *Gallus gallus*, *Lynx lynx*, *Meleagris gallopavo f. domestica*, *Nasua nasua*, *Oryx beisa*, *Puma concolor*, and *Speothos venaticus*. Table 5 shows the content of individual B vitamins in the analysed blood samples of selected animals and human. The animals can be grouped according to their taxa and feeding strategy into i) fish, ii) herbivores, iii) omnivores, iv) carnivores, and v) birds. The fish were grown and artificially fed in experimental fishponds in controlled conditions which may explain relatively high amount of detected B vitamins in their blood samples.

Vitamin B1 was detected in some of the sampled herbivores, fish and birds, whereas no B1 was found in any carnivore. In human whole blood, the reference range of vitamin B1 is 2.5–7.5 µg/L [26].

Vitamin B2 was present in all examined vertebrates. As mentioned above, significant differences can be seen between fish and the rest of the sampled animals, with up to two order higher content of B2 in fish. The levels of vitamin B2 in human blood have been previously reported as 40–240 µg/L [36, 37].

Nicotine acid, a form of vitamin B3, was not detected in any sample, while nicotinamide, another form of B3, was present in all the tested samples. In feline carnivores, the level of nicotinamide was significantly lower compared to all the other groups. The level of nicotinamide in humans was reported in the range of 2300–6500 µg/L in blood and 620–1700 µg/L in serum [26].

Vitamin B5 was detected in all animals except for *Puma* and *Oryx*. Similarly to vitamin B3, vitamin B5 levels were lower in carnivores than in most other animals studied, with the

Table 5. Content of analysed B vitamins in various different animal species.

| Group | Latin name | English name | Age | Sex | Thiamine (B1) | Riboflavin (B2) | Niacin (B3) | Niacinamid (B3-AM) | Pantothenic acid (B5) | Pyridoxine (B6) | 4-Pyridoxic acid (B6-4PA) | Biotin (B7) | Cyanoco-balamine (B12) |
|---|---|---|---|---|---|---|---|---|---|---|---|---|---|
| | | | | | | | | Concentration[a] in whole blood [µg/l] | | | | | |
| FISH | *Acipenser ruthenus* | Sturgeon | 7 | Male | 4.30 ± 0.41 | 768 ± 86 | ND | 45400 ± 3500 | 488.3 ± 3.2 | ND | ND | ND | ND |
| | *Acipenser ruthenus* | Sturgeon | 4 | Female | ND | 1211 ± 35 | ND | 12150 ± 670 | 367.3 ± 2.7 | ND | ND | ND | ND |
| | *Acipenser ruthenus* | Sturgeon | 8 | Male | 20.2 ± 1.2 | 1300 ± 1000 | ND | 18550 ± 240 | 658 ± 23 | ND | ND | ND | ND |
| | *Acipenser ruthenus* | Sturgeon | 7 | Female | 11.47 ± 0.78 | 1400 ± 340 | ND | 8330 ± 210 | 714 ± 18 | ND | ND | ND | ND |
| | *Acipenser ruthenus* | Sturgeon | 7 | Female | ND | 1840 ± 430 | ND | 35800 ± 5200 | 493.3 ± 5.9 | ND | ND | ND | ND |
| HERBOVIRES | *Alces alces* | European Moose | 2 | Male | 5.77 ± 0.60 | 19.64 ± 0.78 | ND | 9720 ± 760 | 141.5 ± 7.9 | ND | 5.73 ± 0.46 | ND | ND |
| | *Bison bonasus* | Aurochs | 1,5 | Female | ND | 17.9 ± 1.7 | ND | 3120 ± 260 | 359 ± 12 | ND | ND | ND | ND |
| | *Camelus bactrianus* | Bactrian Camel | 17 | Male | 5.61 ± 0.68 | 36.83 ± 0.83 | ND | 9400 ± 770 | 23.50 ± 0.15 | ND | 7.32 ± 0.42 | ND | ND |
| | *Dama dama* | European Fallow Deer | 5 | Male | 6.1 ± 1.1 | 32.88 ± 0.83 | ND | 7350 ± 380 | 71.5 ± 1.4 | ND | 10.10 ± 0.31 | ND | ND |
| | *Elephas maximus* | Indian Elephant | 46 | Female | ND | 6.3 ± 1.1 | ND | 293 ± 17 | 10.78 ± 0.31 | ND | ND | ND | ND |
| | *Elephas maximus* | Indian Elephant | 63 | Female | ND | 5.08 ± 0.50 | ND | 2219.8 ± 6.0 | 13.1 ± 1.0 | ND | ND | ND | ND |
| | *Equus przewalskii* | Przewalski's Horse | 16 | Female | ND | 9.5 ± 1.0 | ND | 4300 ± 260 | 90.6 ± 6.1 | ND | ND | ND | ND |
| | *Equus przewalskii* | Przewalski's Horse | 1 | Male | ND | 7.26 ± 0.90 | ND | 7180 ± 630 | 218 ± 12 | ND | ND | ND | ND |
| | *Equus przewalskii* | Przewalski's Horse | 1 | Male | ND | 7.0 ± 2.3 | ND | 3480 ± 340 | 132.8 ± 4.1 | ND | 7.31 ± 0.43 | ND | ND |
| | *Equus przewalskii* | Przewalski's Horse | 16 | Female | ND | 3.49 ± 0.42 | ND | 4700 ± 520 | 80.1 ± 2.8 | ND | ND | ND | ND |
| | *Fukomys mechowi* | Giant Grouse | 12 | Male | 36.6 ± 5.8 | 81.8 ± 7.1 | ND | 1263 ± 35 | 650 ± 57 | ND | 20.1 ± 4.5 | ND | ND |
| | *Oryx beisa* | Beisa Rhinoceros | 22 | Female | ND | 23.1 ± 3.3 | ND | 3661 ± 26 | ND | ND | 15.05 ± 0.59 | ND | ND |
| OMNIVORES | *Nasua nasua* | Brown-Nosed Coati | 10 | Male | ND | 83.4 ± 4.7 | ND | 1249 ± 21 | 22.2 ± 2.3 | ND | ND | ND | ND |
| CARNIVORES | *Arctocephalus pusillus* | South African Sea Lion | 19 | Male | ND | 15.70 ± 0.54 | ND | 1795 ± 37 | 8.45 ± 0.71 | ND | ND | ND | ND |
| | *Lynx lynx* | Eurasian Lynx | 2 | Male | ND | 15.8 ± 1.0 | ND | 98.0 ± 7.3 | 15.73 ± 0.46 | 16.63 ± 0.01 | ND | ND | ND |
| | *Lynx lynx* | Eurasian Lynx | 2 | Male | ND | 19.94 ± 0.49 | ND | 81.6 ± 2.6 | 13.64 ± 0.33 | 16.62 ± 0.73 | ND | ND | ND |
| | *Puma concolor* | Cougar | 7 | Female | ND | 9.60 ± 0.24 | ND | 28.9 ± 1.0 | ND | ND | ND | ND | ND |
| | *Speothos venaticus* | Bush dog | 11 | Female | ND | 38.1 ± 3.9 | ND | 7230 ± 230 | 27.5 ± 0.81 | ND | ND | ND | ND |

*(Continued)*

**Table 5.** (Continued)

| Group | Latin name | English name | Age | Sex | Thiamine (B1) | Riboflavin (B2) | Niacin (B3) | Niacinamid (B3-AM) | Pantothenic acid (B5) | Pyridoxine (B6) | 4-Pyridoxic acid (B6-4PA) | Biotin (B7) | Cyanocobalamine (B12) |
|---|---|---|---|---|---|---|---|---|---|---|---|---|---|
| | | | | | | | | Concentration[a] in whole blood [µg/l] | | | | | |
| BIRDS | *Delichon urbicum* | House Martin | n. a.[b] | n.a.[b] | 10.6 ± 2.3 | 59.2 ± 4.9 | ND | 26000 ± 1900 | 446 ± 22 | ND | ND | ND | ND |
| | *Gallus gallus* | Domestic Fowl | 3 | Male | ND | 10.3 ± 1.2 | ND | 5260 ± 480 | 217 ± 20 | ND | ND | ND | ND |
| | *Meleagris gallopavo* | Turkey | 2 | Male | 5.37 ± 2.3 | 12.13 ± 0.46 | ND | 15840 ± 470 | 110.47 ± 0.52 | ND | ND | ND | ND |
| HUMAN | *Homo sapiens sapiens* | Human | 37 | Male | 16.54 ± 0.22 | 27.27 ± 0.59 | ND | 3710 ± 31 | 108.7 ± 1.4 | ND | 31.56 ± 0.90 | 10.9 ± 1.1 | ND |

[a] Expressed as mean ± standard deviation.
[b] Mixed pool obtained from 20 different individuals.

exception of elephants and omnivores (Brown-Nosed Coati). The level of B5 in human blood is reported in rather wide range of 30–400 μg/L [26].

Vitamin B6, pyridoxine, was detected only in two samples of lynx. The catabolic product of vitamin B6, the 4-pyridoxic acid (4PA), was found only in a few analysed herbivores. Concentrations of more than 30 nmol/L (that corresponds to about 5 μg/L) have been traditional indicators of adequate vitamin B6 (pyridoxal phosphate) status in adult humans [15, 38].

No vitamin B7 and B12 was detected in the blood of any of the sampled animal. In the human sample, only B7 was present. The levels of these two B vitamins in humans are reported at low levels in a range of μg/L or ng/L, respectively [15].

## Conclusions

The aim of this work was to quantify simultaneously seven B vitamins and two B vitamin derivatives in a whole blood. The suitability of two different protein precipitating agents was compared using ZnSO4/methanol and trichloroacetic acid. Attention was focused on detailed optimization of mass detection using linear ion trap mass analyser for rapid and robust LC-MS/MS method. The usage of $ZnSO_4$/methanol instead of trichloroacetic acid for protein precipitation step provides high percentages of accuracy, precision and process efficiency. As far as we know, this is the first method that allows to quantify thiamine, riboflavin, niacin, niacinamide, pantothenic acid, pyridoxine, 4-pyridoxic acid, biotin and cyanocobalamine in a whole blood simultaneously. The method was tested on a sample set of several vertebrates showing differences in B vitamin content in dependence on species or their feeding strategies.

## Supporting information

**S1 File. Calibration curves.**
(PDF)

## Author Contributions

**Conceptualization:** David Kahoun, František Vácha, Václav Hypša.

**Formal analysis:** Marie Čížková.

**Funding acquisition:** František Vácha, Václav Hypša.

**Investigation:** David Kahoun, Pavla Fojtíková.

**Methodology:** David Kahoun.

**Resources:** Roman Vodička.

**Validation:** David Kahoun.

**Writing – original draft:** David Kahoun.

**Writing – review & editing:** Pavla Fojtíková, František Vácha, Eva Nováková.

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
