## [Decision Letter · Decision Letter 0]

2 Jun 2022

PONE-D-22-10162Development and validation of an LC-MS/MS method for determination of B vitamins and some its derivatives in whole bloodPLOS ONE

Dear Dr. Kahoun,

Thank you for submitting your manuscript to PLOS ONE. After careful consideration, we feel that it has merit but does not fully meet PLOS ONE’s publication criteria as it currently stands. Therefore, we invite you to submit a revised version of the manuscript that addresses the points raised during the review process.

We look forward to receiving your revised manuscript.

Kind regards,

Joseph Banoub, Ph,D., D. Sc.

Academic Editor

PLOS ONE

Journal Requirements:

Whilst you may use any professional scientific editing service of your choice, PLOS has partnered with both American Journal Experts (AJE) and Editage to provide discounted services to PLOS authors. Both organizations have experience helping authors meet PLOS guidelines and can provide language editing, translation, manuscript formatting, and figure formatting to ensure your manuscript meets our submission guidelines. To take advantage of our partnership with AJE, visit the AJE website (http://aje.com/go/plos) for a 15% discount off AJE services. To take advantage of our partnership with Editage, visit the Editage website (www.editage.com) and enter referral code PLOSEDIT for a 15% discount off Editage services.  If the PLOS editorial team finds any language issues in text that either AJE or Editage has edited, the service provider will re-edit the text for free.

A clean copy of the edited manuscript (uploaded as the new *manuscript* file).

Reviewers' comments:

Reviewer's Responses to Questions

**Comments to the Author**

1. Is the manuscript technically sound, and do the data support the conclusions?

Reviewer #1: Yes

Reviewer #2: Yes

2. Has the statistical analysis been performed appropriately and rigorously? 

Reviewer #1: Yes

Reviewer #2: N/A

3. Have the authors made all data underlying the findings in their manuscript fully available?

Reviewer #1: Yes

Reviewer #2: No

4. Is the manuscript presented in an intelligible fashion and written in standard English?

Reviewer #1: Yes

Reviewer #2: No

5. Review Comments to the Author

Reviewer #1: The manuscript by David Kahoun et al. is an interesting LC-MS/MS study for quantification of various vitamins B in whole blood.

The work is well-structured, well-written and easy to understand. However the references are not sufficiently updated.

Authors could improve the introduction section citing some recent studies for the analysis of various matrices: (these are only some receent examples)

• Roelofsen-de Beer RJAC, van Zelst BD, Wardle R, Kooij PG, de Rijke YB. Simultaneous measurement of whole blood vitamin B1 and vitamin B6 using LC-ESI-MS/MS. J Chromatogr B Analyt Technol Biomed Life Sci. 2017 Sep 15;1063:67-73. doi: 10.1016/j.jchromb.2017.08.011. Epub 2017 Aug 12. PMID: 28846867.

• Nguyen QH, Hoang AQ, Truong TMH, Dinh TD, Le TT, Luu THT, Dinh VC, Nguyen TMT, Vu TT, Nguyen TAH. Development of Simple Analytical Method for B-Group Vitamins in Nutritional Products: Enzymatic Digestion and UPLC-MS/MS Quantification. J Anal Methods Chem. 2021 May 5;2021:5526882. doi: 10.1155/2021/5526882. PMID: 34035973; PMCID: PMC8116160.

Thus, I would recommend the publication of the manuscript with minor revisions

Reviewer #2: In the manuscript titled “Development and validation of an LC-MS/MS method for determination of B vitamins and some its derivatives in whole blood”, Kahoun et al. present a LC-MS/MS based mass spectrometry method to examine multiple B-vitamins (and derivatives) within vertebrate samples. Specific comments are as follows:

1. Please provide a data supplement to show the calibration curves for each examined B vitamin (and derivative).

2. While it appears that the zinc sulfate-based precipitation was more advantageous, a figure showing the direct comparison of the method vs. TCA is missing. Please provide plots showing the correlation of each B vitamin (and derivatives) between both methods.

3. For Table 5, provide information for age, sex, and n-value.

4. Some modest changes to language are needed.

6. PLOS authors have the option to publish the peer review history of their article (what does this mean?). If published, this will include your full peer review and any attached files.

Reviewer #1: No

Reviewer #2: No

---

## [Author Response · Author response to Decision Letter 0]

29 Jun 2022

Response to reviewers' comments - manuscript: PONE-D-22-10162

Reviewer #1: The manuscript by David Kahoun et al. is an interesting LC-MS/MS study for quantification of various vitamins B in whole blood. The work is well-structured, well-written and easy to understand. However, the references are not sufficiently updated.:

1. Authors could improve the introduction section citing some recent studies for the analysis of various matrices: (these are only some receent examples)

• Roelofsen-de Beer RJAC, van Zelst BD, Wardle R, Kooij PG, de Rijke YB. Simultaneous measurement of whole blood vitamin B1 and vitamin B6 using LC-ESI-MS/MS. J Chromatogr B Analyt Technol Biomed Life Sci. 2017 Sep 15;1063:67-73. doi: 10.1016/j.jchromb.2017.08.011. Epub 2017 Aug 12. PMID: 28846867.

• Nguyen QH, Hoang AQ, Truong TMH, Dinh TD, Le TT, Luu THT, Dinh VC, Nguyen TMT, Vu TT, Nguyen TAH. Development of Simple Analytical Method for B-Group Vitamins in Nutritional Products: Enzymatic Digestion and UPLC-MS/MS Quantification. J Anal Methods Chem. 2021 May 5;2021:5526882. doi: 10.1155/2021/5526882. PMID: 34035973; PMCID: PMC8116160.

Authors: The introduction section has been updated with recent studies. Thank you for the notice.

Reviewer #2: In the manuscript titled “Development and validation of an LC-MS/MS method for determination of B vitamins and some its derivatives in whole blood”, Kahoun et al. present a LC-MS/MS based mass spectrometry method to examine multiple B-vitamins (and derivatives) within vertebrate samples. Specific comments are as follows:

1. Please provide a data supplement to show the calibration curves for each examined B vitamin (and derivative).

Authors: The required data supplement was added (S1 Calibration Curves). Thank you for the important suggestion.

2. While it appears that the zinc sulfate-based precipitation was more advantageous, a figure showing the direct comparison of the method vs. TCA is missing. Please provide plots showing the correlation of each B vitamin (and derivatives) between both methods.

Authors: The required plots were added as Fig 2 and Fig 3. Thank you for improving the visual presentation of our results.

3. For Table 5, provide information for age, sex, and n-value.

Authors: The required information is provided in the Table 5. Thank you for your request to fill in these important missing data.

4. Some modest changes to language are needed.

Authors: The required language changes were made. Thank you for the thorough language check of the text.

---

## [Editor Report · Decision Letter 1]

1 Jul 2022

Development and validation of an LC-MS/MS method for determination of B vitamins and some its derivatives in whole blood

PONE-D-22-10162R1

Dear Dr. Kahoun,

We’re pleased to inform you that your manuscript has been judged scientifically suitable for publication and will be formally accepted for publication once it meets all outstanding technical requirements.

Kind regards,

Joseph Banoub, Ph,D., D. Sc.

Academic Editor

PLOS ONE
---

## [Editor Report · Acceptance letter]

7 Jul 2022

PONE-D-22-10162R1 

Development and validation of an LC-MS/MS method for determination of B vitamins and some its derivatives in whole blood 

Dear Dr. Kahoun:

I'm pleased to inform you that your manuscript has been deemed suitable for publication in PLOS ONE. Congratulations! Your manuscript is now with our production department. 

Kind regards, 

on behalf of

Dr. Joseph Banoub 

Academic Editor

PLOS ONE